# Telemedicine in Neuro-Oncology—An Evaluation of Remote Consultations during the COVID-19 Pandemic

**DOI:** 10.3390/cancers15164054

**Published:** 2023-08-11

**Authors:** Jonas Feldheim, Teresa Schmidt, Christoph Oster, Julia Feldheim, Martin Stuschke, Walter Stummer, Oliver Grauer, Björn Scheffler, Carsten Hagemann, Ulrich Sure, Christoph Kleinschnitz, Lazaros Lazaridis, Sied Kebir, Martin Glas

**Affiliations:** 1Division of Clinical Neuro-Oncology, Department of Neurology and Center for Translational Neuro- and Behavioral Sciences (C-TNBS), University Medicine Essen, University Duisburg-Essen, 45147 Essen, Germany; 2Section Experimental Neurosurgery, Department of Neurosurgery, University Hospital Würzburg, 97080 Würzburg, Germany; 3German Cancer Consortium (DKTK), Partner Site University Medicine Essen, 45147 Essen, Germany; 4DKFZ-Division Translational Neuro-Oncology, West German Cancer Center (WTZ), DKTK Partner Site, University Medicine Essen, University Duisburg-Essen, 45147 Essen, Germany; 5Department of Neurosurgery, University Hospital Essen, 45147 Essen, Germany; 6Department of Radiation Oncology, University Hospital Essen, 45147 Essen, Germany; 7Department of Neurosurgery, University Hospital Münster, 48149 Münster, Germany; 8Department of Neurology, University of Münster, 48149 Münster, Germany

**Keywords:** COVID-19, SARS-CoV-2, oncology, cancer, video, consultations, glioma, brain

## Abstract

**Simple Summary:**

The COVID-19 pandemic has driven the expansion of remote consultations in medical care. However, their implementation in neuro-oncology comprises unique challenges and opportunities. Evidence for how the complex patient–doctor relationships and neurological examinations required in the treatment of brain tumor patients can be translated to video consultation is scarce. Therefore, we analyzed over 3700 consultations at our institution in order to determine how patients that decided to participate in remote consultations distinguished from patients that did not. Additionally, we queried them about their reasons, encounters, problems/limitations, opportunities, emotional impact and future directions regarding their experiences with telemedicine during their treatment with an anonymized survey. With this, we are able to provide guidance on suitable patient subgroups, occasions and future directives to expand and adapt telemedicine patient care in neuro-oncology in an individualized and evidence-based concept.

**Abstract:**

In order to minimize the risk of infections during the COVID-19 pandemic, remote video consultations (VC) experienced an upswing in most medical fields. However, telemedicine in neuro-oncology comprises unique challenges and opportunities. So far, evidence-based insights to evaluate and potentially customize current concepts are scarce. To fill this gap, we analyzed >3700 neuro-oncological consultations, of which >300 were conducted as VC per patients’ preference, in order to detect how both patient collectives distinguished from one another. Additionally, we examined patients’ reasons, suitable/less suitable encounters, VC’s benefits and disadvantages and future opportunities with an anonymized survey. Patients that participated in VC had a worse clinical condition, higher grade of malignancy, were more often diagnosed with glioblastoma and had a longer travel distance (all *p* < 0.01). VC were considered a fully adequate alternative to face-to-face consultations for almost all encounters that patients chose to participate in (>70%) except initial consultations. Most participants preferred to alternate between both modalities rather than participate in one alone but preferred VC over telephone consultation. VC made patients feel safer, and participants expressed interest in implementing other telemedicine modalities (e.g., apps) into neuro-oncology. VC are a promising addition to patient care in neuro-oncology. However, patients and encounters should be selected individually.

## 1. Introduction

In the spring of 2020, everyday life worldwide took a drastic turn. A new strain of coronavirus, classified as SARS-CoV-2, explosively spread in a pandemic that would sooner or later affect all countries of the world [1]. Due to a lack of specific treatments or vaccines, management in the initial phase of the pandemic revolved around preventive strategies to reduce viral transmission. This represented an unprecedented challenge for hospitals, already forced to suddenly expand their capacities, to maintain sufficient care for their regular patients with other diagnoses.

However, every challenge contains new opportunities and solutions. Driven by the SARS-CoV-2 pandemic, one of the opportunities that arose was the quick acceleration and promotion of telemedicine [2]. Patient care via telemedicine is not an entirely new concept. Yet, in the last decade, it was mainly implemented as an option to include specialists in emergencies [3] or to provide patients in rural and regional communities with medical care [4]. One typical example of telemedicine is remote consultations via video (VC). As they allow for sustained doctor–patient contact while maintaining social distancing and preventing infection risk, VC were well positioned to address the arising challenges. Consequently, multiple disciplines and institutions expanded their offer and implemented VC [5]. However, while it is easy to believe that the benefit of converting consultation hours to VC might greatly outweigh the disadvantages in some medical fields (e.g., for patients suspected to be infected with SARS-CoV-2), things are less evident in the care of brain tumor patients.

On the one hand, a history of cancer was associated with a higher risk of adverse events in COVID-19 patients [6,7,8], and multiple common tumor treatments can lead to immunodeficiency, making cancer patients a risk group that were advised to minimize hospital visits and elective admissions [9,10]. On the other hand, VC also come with limitations. Frequent and comprehensive neurological examinations that might not translate sufficiently to VC are highly relevant for neuro-oncological patients [11]. In addition, VC have been accused of complicating communication or creating emotional distance between patients and doctors [12,13]. Small physical gestures to comfort the patient or express sympathy might get lost [14]. One may argue that this is particularly troublesome for a neuro-oncological consultation, as the communication in neuro-oncology can be highly emotional and challenging on a personal level, for instance, when breaking the diagnosis of a brain tumor (recurrence) or discussing goals of care. It remains to be seen whether a picture and video transmission can replace the complex doctor–patient relationship in personal contact [14]. Therefore, communication restrictions or additional emotional barriers that arise with VC might also significantly decrease the standard of patient care in neuro-oncology. 

Despite the unclear state and additional hurdles such as technical issues, potential jurisdictional boundaries (e.g., ambiguous regulation of sick certificates) and changing regulations regarding the reimbursement by health insurance, we decided to offer patients treated in our Specialized Outpatient Unit for Neuro-Oncology the alternative to participate in VC instead of face-to-face consultations beginning by the end of March 2020. Since then, we have experienced the opportunities and limitations of telemedicine in neuro-oncology. Though two large treatment facilities have reported some exciting experiences with VC in neuro-oncology [15] and experts of the neuro-oncology section of the American Academy of Neurology published a detailed guide on telemedicine [16], the currently published reports are mainly based on the experience of experts, a preliminary data basis or primarily focused on pediatric neuro-oncology [17,18,19,20,21,22]. To the best of our knowledge, a more detailed evaluation of VC in neuro-oncology has not been provided so far. 

Therefore, we aimed to contribute to an evidence-based evaluation of VC in neuro-oncology with a comprehensive analysis of our patient collective. In this study, we first retrospectively analyzed all consultations in our Specialized Outpatient Clinic in a 15-month timeframe to (1) detect how the patient collective participating in VC distinguished from those that did not. Further, we invited the patients that participated in VC to complete a survey to (2) understand the reasons behind their choice for VC, (3) identify suitable/less suitable encounters for VC, (4) obtain the patients’ view on benefits of VC as well as disadvantages and (5) determine opportunities for improvement or other telemedicine options to improve patients’ care.

## 2. Materials and Methods

### 2.1. Study Design 

In this monocentric trial with retrospective and cross-sectional aspects, we screened all patients that were treated in the Specialized Outpatient Clinic of the Division of Clinical Neuro-Oncology, Department of Neurology, University Medicine Essen, University Duisburg-Essen, Essen, Germany, between the 1 April 2020 and the 30 June 2021. The study was approved by the Institutional Review Board of the University of Duisburg-Essen (reference number: 1-10249-BO).

Patients were included in the trial if at least one physician’s consultation in the Specialized Outpatient Clinic had taken place. We included patients regardless of the diagnosis that led to their consultation, even if the diagnosis was later verified to fall outside the neuro-oncological field. Neuro-oncological patients treated in the Center for Neuro-Oncology of the University Hospital by other departments (e.g., neurosurgery, radiotherapy) or received inpatient treatment without previous/later treatment at the Specialized Outpatient Clinic of the Division of Clinical Neuro-Oncology at least once were excluded from this trial. Additionally, we did not count consultations via our emergency department. VC were offered to all patients as an alternative to face-to-face consultations for every consultation except for the admission of intravenous chemotherapy/tumor therapy. Patients’ preferences for each consultation determined the mode of participation (video vs. in-person). VC were performed with Zava Sprechstunde Online GmbH (https://sprechstunde.online; Essen, Germany; accessed on 30 June 2021). They consisted of video, voice and screen (e.g., to review imaging controls) transmission. 

### 2.2. Collection of Demographic Data

We collected basic demographic and clinical data obtained within the framework of routine clinical assessment. Additionally, we determined the distance of patients’ home addresses to our institution by translating them into latitude and longitude coordinates with OpenStreetMap (© OpenStreetMap; https://www.openstreetmap.org/copyright; accessed on 10 July 2023) and calculated the distance between both points with a version of the haversine formula JavaScript (© Chris Veness; https://www.movable-type.co.uk/scripts/latlong.html; accessed on 10 July 2023) adapted to the use with SPSS statistics 28 (IBM, Armonk, NY, USA). Furthermore, we performed a database query of our clinical information system extracting the number of appointments that included contact with a physician (independent of the reason or extent) divided between contacts made in person or via VC. In the case of multiple consultations in the given timeframe, we extracted the data (e.g., age) from the last documented consultation before 30 June 2021. Similarly, the diagnoses from the last consultation before 30 June 2021 were kept, as we believe that patients’ knowledge about their diagnosis might have influenced their decisions more than the correct classification according to current WHO guidelines. 

Next, we contacted patients that had taken part in a VC at our institution in the chosen timeframe at least once and asked them to answer an anonymized survey regarding their experience with the VC (Appendix A) with SurveyMonkey (www.surveymonkey.com; San Mateo, CA, USA; accessed on 10 July 2023). If patients were deceased or unable to complete the survey alone, close relatives were asked to participate representatively. For reasons of simplicity, we refer to all persons to complete the survey as ’patients’ or participants for most of the manuscript, except for those sections where we observed a misdistribution depending on the group to complete the survey. In these instances, we divide our subgroups into patients that completed the survey by themselves (P), patients that completed the survey with the help of relatives (PR) and surveys that were filled out by relatives alone (R). All data collected were stored and analyzed anonymously. 

### 2.3. Statistical Analyses

We performed statistical analyses with IBM SPSS Statistics 28 (IBM Corporation, Armonk, NY, USA). Data on SARS-CoV-2 infections were extracted via NPGEO Corona (https://npgeo-corona-npgeo-de.hub.arcgis.com; accessed on 10 July 2023) and were based on data provided by the Robert-Koch-Institute, Berlin, Germany. We included each month’s incidence reported on the 15th or the next day after that. Normality was tested with Kolmogorov–Smirnov and Shapiro–Wilk tests, additionally considering the skewness, kurtosis, and a visual representation in a scatter plot. We conducted Spearman’s correlation to investigate associations/correlations (Spearman). In order to compare distribution between groups, we performed the Fisher exact test (Fisher), Chi-squared test (Χ^2^), Mann–Whitney U test (MWU) or Kruskall–Wallis test with post hoc Dunn’s test and Bonferroni correction (KW/D) depending on the groups, variables and objectives.

## 3. Results

### 3.1. Patients with Glioblastoma, Long Travel Distance and Low Karnofsky Were More Likely to Choose VC 

Between 1 April 2020 and 30 June 2021, we included a total of 612 patients with 3751 individual consultations. In the median, these patients had four (quartiles: 2–8) physician consultations at our Specialized Outpatient Unit over the course of 15 months. Of the 3.751 consultations, 313 (8.3%) were conducted via VC. In total, 114 patients chose to participate in VC (18.7%). The gender ratio in our collective was balanced (male: *n* = 310, 50.7%; female: *n* = 302, 49.3%). Patients had a mean Karnofsky index of 80% and were predominantly diagnosed with tumors classified as central nervous system (CNS) World Health Organization (WHO) tumor grade 4. They lived a median distance of 17.2 km (quartiles: 7.1–33.5 km) away from our institution and were in median of 53 years (quartiles: 40–63 years) old at the time of the final appointment in the study’s time frame.

Patients’ choice of VC correlated with the incidence of SARS-CoV-2 infections in Essen at the respective times (Spearman’s Rho: r = 0.5, *p* = 0.03; Figure 1a). However, we identified a general tendency towards more patients participating in VC during the pandemic. For instance, we observed that more patients chose telemedicine during the peak of incidence in the spring of 2021 compared to the winter of 2020, even though the SARS-CoV-2 incidence in Essen was reported to be higher during the peak in the winter of 2020 (Figure 1a).

We wondered whether we could identify subgroups of patients that had more commonly chosen VC over consultations in person and, therefore, compared the collective of patients who had never taken part in a video consultation with those who had taken part at least once. As all of the metric variables included were verified not to be normally distributed (Kolmogorov–Smirnov: *p* < 0.01 for all metric variables; Shapiro–Wilk: *p* < 0.01 for all metric variables), we chose non-parametric tests to compare both groups, as well as Fisher exact and Chi-squared tests for ordinal and nominal variables.

While patients’ sex and age were distributed equally between both groups, patients with CNS WHO grade 4 tumors (70.2% vs. 42.6%), such as glioblastoma (69.4% vs. 39.4%), were more likely to participate in a VC. Additionally, we observed a significant tendency towards VC when patients had a lower Karnofsky index (Figure 1b) and a longer distance to our institution (median distance 37.5 km vs. 14.4 km). The comparison between both groups is summarized in Table 1.

Given are the absolute numbers or the median, with the percentage of the subgroup or quartiles in brackets. The abbreviations in brackets after the *p*-value indicate the statistical test used: Fisher, Fisher’s exact test; MWU, Mann–Whitney U test; Χ^2^, Chi-squared test. Additional abbreviations: IDHwt, Isocitrate dehydrogenase wild-type. IDHmut, Isocitrate dehydrogenase mutated. Diagnoses were not updated according to the new WHO classification or developments after 30 June 2021 in order to correctly reflect the information that builds patients’ perception. 

### 3.2. Ratio of VC Was Associated with the Travel Distance and the Number of Consultations in Total

Next, we wondered whether the demographic and clinical attributes we identified to be associated with the decision to participate in VC might also influence the ratio of VC to ‘classical’ face-to-face consultations in the subgroup of patients that participated in telemedicine at least once. In this subgroup of 114 patients, we observed a significant correlation between the distance that patients had to travel for an appointment in person (r = 0.70, *p* < 0.01, Figure 2a) and the ratio of VC and the number of consultations in total (r = −0.76, *p* < 0.01, Figure 2b). Patients’ Karnofsky, CNS WHO grade (both Spearman, *p* > 0.05) and diagnosis (Kruskall–Wallis test, *p* = 0.38, post hoc: Dunn’s test with correction according to Bonferroni, all subgroups *p* > 0.05) were not associated with the percentage of VC in the subgroup of patients that participated at least once. Consequently, the subset of patients that clearly chose VC (>75% of the consultations in total) over appointments in person participated in only a median of three consultations (quartiles 1–5 consultations) in total, lived a median distance of 173 km (quartiles 80–266 km) away from our institution and was predominantly diagnosed with glioblastoma or gliosarcoma CNS WHO grade 4 (79.5%). 

### 3.3. VC Were Considered an Adequate Alternative for Most Occasions but Not for the Initial Consultation

While the statistical associations might provide some insight, we were curious about patients’ reasons behind their choices and conclusions after experiencing the VC. Therefore, we contacted all 114 patients that had participated in a VC at least once and invited them to complete an anonymized survey (Appendix A). We received 58 responses (response rate: 50.9%). Seventeen (P, 29.3%) of the surveys were completed by patients themselves, 19 (PR, 32.8%) with the help of relatives and 22 (R, 37.9%) only by relatives and without the patient (e.g., if the patient was deceased). Congruently with the previously obtained information, most patients (40/69.9%) suffered from a glioblastoma CNS WHO grade 4. Thirty-three (56.9%) of the patients were male and 25 (43.1%) were female. Their median age was 55 years (quartiles 44.5–64 years), and the median distance from residence to our institution was stated to be between >50–100 km. Eighteen (31.0%) of the patients had only participated in one VC, 13 in two (22.4%) and 27 (46.6%) patients in more than two VC. 

The most common factors influencing patients’ decision to participate in VC were comfort (*n* = 35, 60%), the travel distance (*n* = 30, 52%) and the aim to avoid infections with SARS-CoV-2 (*n* = 28, 48%), while only a minority based their decision on a poor clinical condition (*n* = 15, 26%) or insufficient possibilities of transportation to the hospital (*n* = 6, 10%; multiple choice question, Figure 3a). However, both reasons were unequally present in the subgroups divided by the person to fill out the survey. While 0/17 (P, 0%) patients claimed that a poor clinical state influenced their decision, 4/19 (PR, 21%) patients that required help from their relatives and 11/22 (R, 50%) of relatives that had to fill out the survey representatively did so (*p* = 0.02, X^2^, Figure 3b). Opposite to this, only 1/22 (5%) of surveys that relatives filled out without patients and 0/19 (0%) of those completed with the help of a relative gave insufficient possibilities of transportation to the hospital as a reason, whereas 5/17 (11%) of patients that completed the survey alone did (*p* = 0.08, X^2^, Figure 3b). Additionally, only 1/40 (3%) of these patients suffered from glioblastoma, while 5/12 (42%) participants that gave insufficient possibilities of transportation as a reason were diagnosed with other diseases/tumor types (*p* < 0.01, X^2^). Unsurprisingly, patients that gave long travel distances as a reason to participate in VC lived significantly further away from Essen (*p* < 0.01, MWU). 

Next, we asked patients what the occasion/topics discussed for their consultation had been and for which encounter they would hypothetically rate the VC as a fully adequate alternative, regardless of whether they experienced a VC for this occasion. In reality, the most common encounters were the reviews of MRI controls (*n* = 50, 86%), followed by the control visit before the start of a new cycle of chemotherapy (*n* = 22, 38%), determining the next steps after diagnosis of a tumor relapse (*n* = 21, 36%), advice about clinical trials (*n* = 16, 28%), a consultation due to a clinical deterioration (*n* = 14, 24%), obtaining a second opinion (*n* = 11, 19%), and the initial consultation (*n* = 7, 12%). 

Interestingly, more patients considered VC a fully adequate alternative before starting a new cycle of chemotherapy (*n* = 36, 62%), getting advice about clinical trials (*n* = 34, 59%) or obtaining a second opinion (*n* = 24, 41%) than had actually participated in a VC for these occasions. Patients’ assessments of the feasibility of video consultations regarding the next steps after diagnosis of a tumor relapse (*n* = 24, 41%), clinical deterioration (*n* = 21, 36%) or the initial consultations (*n* = 8, 14%) closely resembled the numbers/percentages of the consultations that had actually taken place (Figure 3c).

Additionally, we were interested in whether patients considered the encounters that they had actually experienced via VC to be a fully adequate alternative to consultations in person. Notably, except from initial consultations (43%), at least 70% of patients from each group rated VC for the occasions they had experienced in reality as a fully adequate alternative to face-to-face consultations. Video consultations to obtain a second opinion or advice about clinical trials were even rated fully adequate by 100% of those who experienced them (Figure 3d). 

Interestingly, relatives (12/22, 55%) that completed the survey representatively for the patients considered VC significantly more viable in the event of a clinical worsening than patients alone (6/17, 35%) or with a relative (3/16, 19%; *p* = 0.03, X^2^). Fifty-seven per cent (*n* = 33/58) of our patients stated not to have made any negative experiences with the VC, while 26% (15/58) experienced technical issues, 12% (7/58) found some of the physical and neurological examinations too limited/superficial and 9% (5/58) rated the discussion atmosphere as too impersonal. Interestingly, none of our patients (0/58) expressed data protection/privacy concerns. 

### 3.4. VC Had a Significantly Lower Expense yet Were Found to Be Slightly Less Personal Than Face-to-Face Consultations

Fifty-two out of fifty-eight (89.7%) of the patients that completed the survey stated that they would prefer the offer to continue independently of the COVID-19 pandemic. However, almost all participants considered a concept with alternating consultations via video and in person the most sensible. Only 2/58 patients (3.4%) preferred to participate solely in VC. Twenty-four patients (24/58, 41.4%) supported predominant participation in VC with few consultations in person in between, while 14/58 (24.1%) preferred a balanced ratio, and 18/58 (31.0%) tended towards face-to-face consultations with occasional VC. None of the patients preferred a concept without VC. 

Before the rise of VC, remote consultations were almost exclusively performed via telephone. Therefore, we asked the participants to compare both modalities for us. Forty-five (45/58, 77.6%) participants clearly preferred VC, while 4/58 (6.9%) stated a slight preference, 8/58 (13.8%) had no clear favorite, and one participant (1/58, 1.7%) largely favored telephone calls. Next, we wondered how the participants experienced the different modalities of consultation in comparison. The personal contact of ‘classical’ consultations was in median rated as ‘very good’ (=‘1’) and thereby significantly better than that of video (‘good’ = ‘2’) and telephone consultations (‘satisfactory’ = ‘3’; *p* < 0.01; KW/D, Figure 4a). However, personal contact via video consultations was assessed to be significantly better than via telephone (*p* < 0.01; KW/D). Regarding the expenses for a consultation, patients found video and telephone to be superior to face-to-face consultations, who were only rated as ‘sufficient’ (=‘4’; *p* < 0.01; KW/D; Figure 4b). This assessment was independent of whether relatives or patients completed the survey (*p* > 0.05; KW/D). 

### 3.5. The Offer of VC Made Patients Feel Safer

In advance of the project, multiple patients reported that the additional offer of VC positively impacted them. Therefore, we asked our study participants whether they complied with statements about potential emotional effects and reasons for those. Only half of the participants agreed that the offer of VC lessened their emotional burden, while the other half disagreed. However, most participants claimed that the offer made them feel safer (70%) and better taken care of (76%). When asked about potential reasons, more than 80% of participants consented that the possibility to chose VC, when required, made them feel safer because it was easier to participate in consultations in shorter intervals, easier to consult a specialist in addition to their local physicians and because they had the option of VC in case their clinical state would not allow a consultation in person (Figure 5).

### 3.6. Patients Reported Interest in Implementing Further Telemedicine Modalities in Their Treatment

Finally, we questioned participants about their needs and potential deficiencies in the counselling and treatment process to identify issues that an expansion of telemedicine options could beneficially address. Most participants reported that they were already fully or partially satisfied with the information provided by their general practitioners (45/52, 87%), local specialists (e.g., neurologist, oncologist; 24/34, 71%), neurosurgeon (35/47, 74%), radiation therapists (36/47, 77%), and neurooncologist (52/55, 95%). Regardless of whether it was adequately met during the treatment, participants partially or fully agreed that they felt the need for the following (in descending order): an overview about the treatment and next steps (51/56, 91%), access to commonly understandable information about their disease (50/55, 90%), easy and fast consultation of a specialist (e.g., via VC, 48/55, 87%), automatized documentation of clinical data with notifications if threshold values are exceeded (e.g., laboratory results, 39/49, 80%), accessible possibility to obtain a second opinion (e.g., via VC, 32/44, 73%), access to general information about how to receive social support (e.g., home help, rehabilitation, disability; 37/52, 71%), access to experimental therapies and clinical trials (35/50, 70%), telemedicinic consultations of a psycho-oncologist (25/46, 54%), an exchange/contact with other patients in a similar situation (e.g., via a digital self-help group; 21/49, 43%), and educational videos about supporting measures (e.g., nutrition counselling or autogenic training; 18/49, 37%).

Almost all participants stated a strong interest in using other supplementary telemedicine options (e.g., apps, software, online platform) if the service would be provided to them (50/55, 91%). According to their answers, digital documentation of relevant clinical data with automatic notifications (e.g., in the event of a thrombo-/neutropenia) would be applied by almost all participants of our survey (56/57, 98%). If an app/software had met their requirements/needs, 27 of our participants (27/46, 59%) stated they would have even been willing to pay for such a service. However, most participants believed that treatment via VC (53/53, 100%) or additional telemedicine support for the treatment (e.g., via an app; 51/53, 96%) should be covered by governmental health insurance.

## 4. Discussion

Not unexpectedly, patients’ choice to participate in VC correlated with the incidence of SARS-CoV-2 infections in Essen, once again verifying that the COVID-19 pandemic was a significant driver for the growth of telemedicine [23,24]. However, over time, the percentage of patients that chose to participate in VC rose compared to the incidence rate. As patients, especially those less experienced with modern technologies, reported initial concerns and restraints about technical practicability [8,13,25], this trend is likely associated with the population’s rise in acceptance, experience and adequate specialized equipment for web and video conferencing throughout the pandemic [15,26].

Given that the younger generation is more likely to use new technologies [27], it was slightly surprising that patients’ age appeared to have no impact on the acceptance of VC in our collective. Though patients’ relatives and their familiarity with VC might have played a role, this indicates that VC in neuro-oncology is not only a viable option for young people. Compared to patients that never participated in VC, the collective of patients that participated at least once consisted of a high percentage of glioblastomas, CNS WHO grade 4 tumors and patients with a poor clinical state/low Karnofsky index. Though we cannot provide a definitive conclusion as to why, it is noteworthy that this collection comprises some of the more severely impaired patients treated at our institution. One might speculate that their clinical state played a significant role in participation, as they avoided burdensome travel and could diminish their risk for SARS-CoV-2 infections. Additionally, most glioblastoma patients receive an intense therapeutic regime with regular appointments and might therefore have chosen to avoid travelling to our institution every time by occasionally participating in VC. Additionally, surveys that were filled out by or with the help of patients’ relatives reported significantly more often that a poor clinical state influenced their decision towards VC. Relatives were asked to fill out or help with the survey if patients themselves were unable to or were deceased. This leads us to believe that there is indeed a subgroup of severely impaired patients, mainly diagnosed with glioblastoma, whose main reason for participation in VC was that consultation in person might have been difficult or impossible to achieve otherwise.

As expected, patients that participated in VC lived significantly further away from our institution. One of the main incentives to implement telemedicine has traditionally been to provide adequate medical service for people in remote communities [4]. Even in densely populated areas, specialists for neuro-oncology are still rare, and patients have to expect a long travel time [16]. Consequently, we also observed that patients with long travel distances predominately participated in VC and had fewer consultations in total. These patients classify another subgroup of our collective that most likely chose VC in order to consult a specialist distant from their residence for single or long-interval consultations, presumably additional to their local patient care. This deserves to be noted, as long distances with demanding travel to reach adequate treatment facilities have led to inappropriate treatment, a worse prognosis, and a worse quality of life in cancer patients [28,29,30,31]. One might hypothesize that offering VC could positively impact treatment prognosis and quality of life for this subgroup of patients.

In contrast, patients living within a 50 km radius participated mainly in occasional VCs as an alternative to regular in-person visits. This was especially common for patients with >10 consultations in total. As the most prevalent reason for VC given in our survey was ‘comfort’, we suspect VC offered a welcome relief for patients that had to participate in regular short-interval follow-up visits. Indeed, previous investigations show that follow-up visits via VC are a promising and accepted field of application and neither undermine patients’ satisfaction nor the clinical outcome [15,21,22,32].

Patients and caregivers experienced various types of visits with different requirements during neuro-oncological treatment. In their practice guideline, Strowd, et al. [16] provided an overview of encounters that they considered more or less suitable for telemedicine on general considerations. Our survey participants mostly agreed with their judgement that the monitoring and initiation of chemotherapy and remote surveillance visits after MRI control present highly suitable encounters for VC, as well as second opinion encounters and consultations about clinical trials. Interestingly, urgent consultations due to clinical deterioration were considered more viable when relatives completed the survey alone. This might obviously indicate a difference in opinion between patients and relatives. However, as stated above, relatives were asked to complete the study representatively if the patients were deceased or severely impaired. Therefore, this group also represents the participants who most likely experienced a sudden clinical deterioration due to the brain tumor, which might have shaped their opinion.

Reassuringly, the large majority of patients evaluated the VC encounters that they had actually experienced to be a fully adequate alternative to consultations in person. Therefore, we hypothesize that VC might provide a satisfactory alternative regardless of the encounter if the individual patients’ initial feeling indicates that they prefer a VC for the specific occasion. VC for the first encounter stand out as an exception to this rule and should be avoided whenever possible, except for patients requesting a second opinion or advice about clinical trials. Yet, we want to point out that these observations are solely based on patients’ satisfaction/subjective judgement. We did not objectively assess if the challenges in communication or examination were met in VC.

Similar to the benefits of VC, potential adverse effects or challenges associated with VCs have also been discussed. Technical challenges, data privacy concerns, communication limitations and physical examination limitations are stated most prominently [14,16,33,34,35]. In our collective, approximately one out of four patients experienced technical difficulties, while only a minority reported an impersonal atmosphere and a superficial examination. Over half of the patients stated that they had not experienced any problems/limitations. Against our expectations, none of our patients expressed concerns about data protection.

While implementing the neurological examination via VC remains an issue, one might argue that its limitations are sometimes overstated. Tele-neurology in the initial assessment of stroke patients has been around for over 20 years [36,37]. Current practices allow for a distinction between stroke and mimics as consistent as on-site examinations [38]. Similarly, the diagnostic accuracy of multiple sclerosis [39,40], movement disorders [41,42,43,44], dementia [45,46,47], as well as the care to patients with headaches [48,49,50] and adults with epilepsy [51] has been described to be similar between remote and in-person consultations. As multiple components of the neurological examination are comparable between subspecialties, we hypothesize that an adaptation of the neurological examination also provides satisfactory results in tele-neuro-oncology [16,52]. This would indeed be conclusive with the fact that only a few of our patients subjectively assessed the examination as superficial and also corresponds to the experiences of Fonkem et al. [15]. The most prominent complaints in our patient collective were based on technical issues; however, it has been shown that those are most common in the implementation phase of VC and acceptance and satisfaction tend to grow with time [53].

Before video conferences were technically feasible, physicians and patients occasionally participated in remote consultations via telephone [54,55,56]. Compared to face-to-face consultations, their lesser expense significantly increased accessibility. However, personal contact and communication limitations mostly restricted their use to single-concern or follow-up visits (e.g., discussion of results) [57,58]. It was unclear where VC situated themselves in that regard. Our patients felt that the quality of personal contact with VC is between face-to-face and telephone consultations. However, the expense of VC was rated almost as low as that of telephone consultations. Additionally, most participants preferred VC over telephone consultations. Arguably, VC comprises most of the benefits of telephone consultations with a few additional assets. Compared to face-to-face visits, VC appear to rather provide value as a complementary offer rather than a replacement. Consistently, most survey participants preferred a mixture of VC and ‘classical’ consultations rather than one of the modalities alone.

Following a brain tumor diagnosis, many patients express suffering from an emotional burden that may significantly impact their quality of life. Though internet-guided self-help could not improve glioma patients’ depressive symptoms [22], it has been reported that remote consultation itself can decrease the burden and improve quality of life [16,59,60,61,62,63,64]. Our patients disagreed that VC lessened their emotional burden. However, the majority consented that the option of VC made them feel safer concerning specific situations that might arise, indicating a positive emotional impact.

While our data certainly provide an evidence-based insight, some limitations should be kept in mind: (a) The retrospective part of our study is mainly observational rather than a randomized trial, and unknown biases influencing patients’ choices cannot be excluded. (b) The survey only presents insight into the opinion of patients that participated in VC, while we can only speculate about the reasoning/opinion of the collective of patients that solely decided on face-to-face participation. (c) The survey presents the subjective assessment of a subgroup rather than an objective comparison and might be biased (e.g., we did not compare the emotional burden or quality of life between both groups using standardized tests). (d) Though with a large collective, the analyses are based on a single center. (e) The main focus is on patient-related observations and the use of VC in neuro-oncology. While we acknowledge that other telemedicine applications, such as virtual tumor boards [65,66] or non-patient-related benefits, such as cost savings [67,68], exist, they were outside our work’s focus. (f) Due to patients’ clinical deterioration or decease, some of the questionnaires were filled out by or with the help of relatives. We checked for statistical anomalies between the groups, but we cannot exclude a bias with complete certainty. In cases where we found a misdistribution, we cannot safely extrapolate whether this misdistribution reflects a different opinion between relatives/patients or different experiences due to a patient’s clinical course. However, we believe that our observations will nevertheless significantly increase the publicly available evidence on the use of VC in neuro-oncology and aid others in adapting and improving their patient care via telemedicine.

## 5. Conclusions

VC were considered a fully adequate alternative to face-to-face consultations except for first encounters if the purpose was not to obtain a second opinion or discuss alternative therapeutic options/clinical trials. Patient subgroups that might be especially suitable for VC are patients with long travel distances seeking an expert opinion, patients in poor clinical condition, glioblastoma patients and patients requiring frequent consultations in person. Most participants preferred to alternate between VC and face-to-face consultations rather than participate in one alone. However, VC may be selected over telephone consultations. VC made patients feel safer, and participants expressed their interest in expanding the offer of telemedicine in neuro-oncology further.

## Figures and Tables

**Figure 1 cancers-15-04054-f001:**
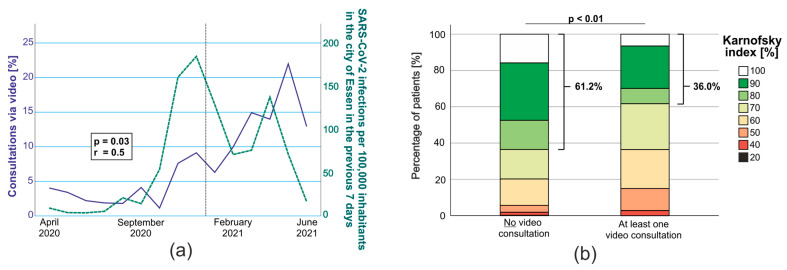
(**a**) Correlation of the ratio of VC at the Specialized Outpatient Clinic for Neuro-Oncology Essen (blue) and the incidence of SARS-CoV-2 infections (green) in the city of Essen (Spearman). (**b**) Comparison of patients’ Karnofsky index between patients who never participated in VC vs. those who did at least once (MWU).

**Figure 2 cancers-15-04054-f002:**
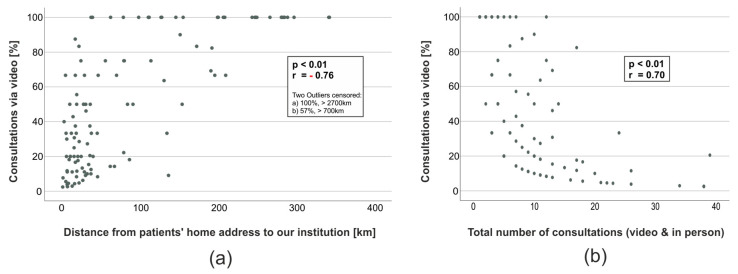
Correlation between distance and total consultations with the ratio of VC. (**a**) Scatter plot and correlation of the ratio of VC out of all consultations and the distance from patients’ residence to our institution (Spearman, two outliers censored from the scatter plot). (**b**) Scatter plot and correlation of the ratio of VC out of all consultations and the total number of consultations via VC and in person (Spearman).

**Figure 3 cancers-15-04054-f003:**
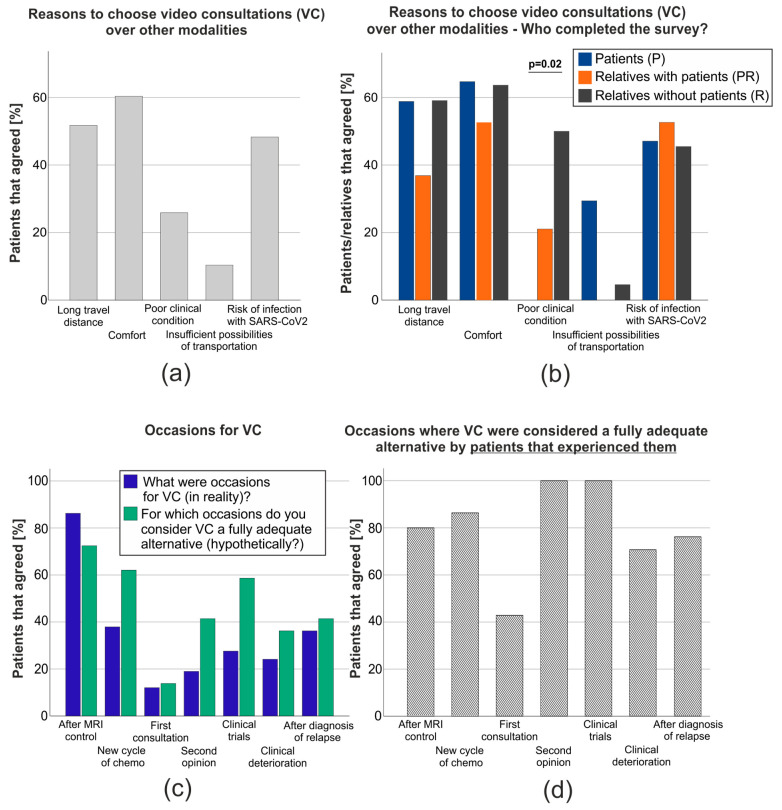
Reasons for and occasions of VC as stated by the participants in our survey. (**a**) Bar graph of participants agreeing/disagreeing with different reasons to choose VC over other modalities (multiple choice question). (**b**) Bar graph of participants agreeing/disagreeing with different reasons to choose VC over other modalities (multiple choice question) divided by the persons completing the survey: patients themselves (P, blue), patients together with their relatives (PR, orange) or relatives alone (e.g., if the patients were unable or deceased, R, black). (**c**) Bar graph of occasions that participants stated to have experienced a VC in reality (blue) and considered a fully adequate alternative to face-to-face consultations irrespective of whether they experienced a VC for this encounter in reality (green, multiple choice questions). (**d**) Participants’ statement of for which occasions where they considered VC to be a fully adequate alternative only including patients/relatives that did in reality experience a VC for the respective occasion (e.g., percentage of patients that considered control visits after the MRI control to be a fully adequate alternative via VC of the collective of patients that experienced at least one control visit after the MRI control as VC).

**Figure 4 cancers-15-04054-f004:**
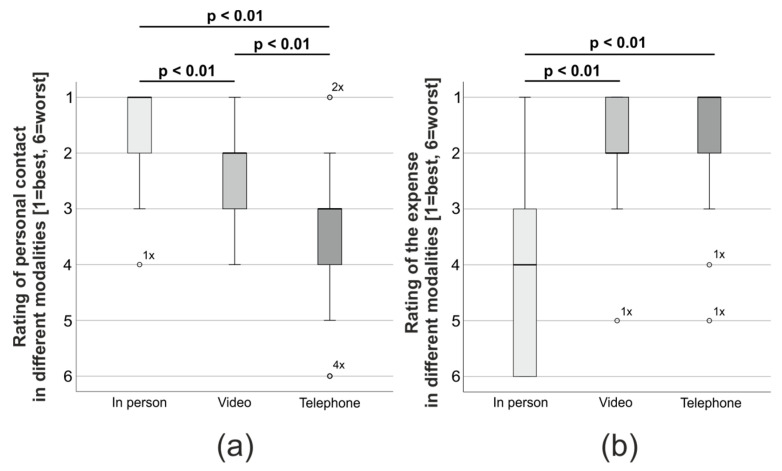
Comparison between participants’ rating of (**a**) personal contact and (**b**) expense between in-person, VC and telephone consultations. The rating is given in the German school grading system (1 = ‘very good’, 2 = ‘good’, 3 = ‘satisfactory’, 4 = ‘sufficient’, 5 = ‘poor’, 6 = ‘insufficient’; KW/D).

**Figure 5 cancers-15-04054-f005:**
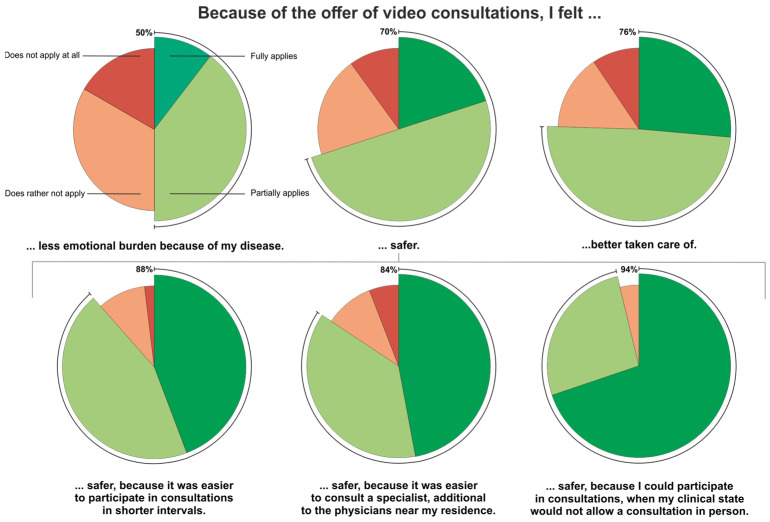
The emotional impact on patients due to the additional offer of VC. Patients were given the statements above and asked whether they ‘fully apply’, ‘partially apply’, ‘do rather not apply’ or ‘do not apply at all’. Pie charts give the percentages of each group.

**Table 1 cancers-15-04054-t001:** Comparison of demographic and clinical characteristics between the patient collective that never participated in VC vs. those that did at least once.

	Patients That Never Participated in VC (*n* = 498)	Patients That Participated in VC at Least Once (*n* = 114)	*p*-Value
Sex	Male: 247 (49.6%) Female: 251 (50.4%)	Male: 63 (55.3%) Female: 51 (44.7%)	0.16 (Fisher)
Age	52.0 years (39–63 years)	54.5 years (43–62 years)	0.43 (MWU)
Karnofsky index	20%: 1 (0.2%) 40%: 8 (1.6%) 50%: 18 (3.6%) 60%: 70 (14.1%) 70%: 78 (15.7%) 80%: 77 (15.5%) 90%: 152 (30.1%) 100%: 76 (15.3%) Not applicable: 18 (3.6%)	20%: 0 (0%) 40%: 3 (2.6%) 50%: 13 (11.4%) 60%: 23 (20.2%) 70%: 27 (23.7%) 80%: 9 (7.9%) 90%: 25 (21.9%) 100%: 7 (6.1%) Not applicable: 7 (6.1%)	**<0.01** **(MWU)**
CNS WHO grade of malignancy	Grade 1: 23 (4.6%) Grade 2: 96 (19.3%) Grade 3: 97 (19.5%) Grade 4: 212 (42.6%) No CNS WHO grade: 70 (14.1%)	Grade 1: 0 (0%) Grade 2: 9 (7.9%) Grade 3: 14 (12.2%) Grade 4: 80 (70.2%) No CNS WHO grade: 11 (9.6%)	**<0.01** **(Χ²)**
Diagnosis	Glioblastoma/Gliosarcoma: 196 (39.4%) Astrocytoma IDHwt: 31 (6.2%) Astrocytoma IDHmut: 87 (17.5%) Oligodendroglioma: 51 (10.2%) Meningioma: 17 (3.4%) Metastasis: 18 (3.6%) Pilocytic Astrocytoma: 7 (1.4%) Craniopharyngioma: 2 (0.4%) Medulloblastoma: 7 (1.4%) Lymphoma: 2 (0.4%) Ependymoma: 9 (1.8%) Others: 71 (14.2%)	Glioblastoma/Gliosarcoma: 79 (69.2%) Astrocytoma IDHwt: 6 (5.3%) Astrocytoma IDHmut: 11 (9.6%) Oligodendroglioma: 3 (2.6%) Meningioma: 0 (0%) Metastasis: 3 (2.6%) Pilocytic Astrocytoma: 1 (0.9%) Craniopharyngioma: 0 (0%) Medulloblastoma: 0 (0%) Lymphoma: 0 (0%) Ependymoma: 0 (0%) Others: 11 (9.6%)	**<0.01** **(Χ^2^)**
Distance	14.4 km (6.3–25.6 km)	37.5 km (17.2–138.9 km)	**<0.01** **(MWU)**

## Data Availability

Data are contained within the article. Raw data can be obtained from the corresponding authors on reasonable request.

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
