# Peer review of "Telemedicine in Neuro-Oncology—An Evaluation of Remote Consultations during the COVID-19 Pandemic"

_cancers, 2023, doi:10.3390/cancers15164054_

Round 1

Reviewer 1 Report

I appreciate the opportunity to review this work.  This retrospective study evaluates the use of televideo consultation at the peak of the SARS-CoV-2 pandemic in a neuro-oncology practice and surveys patients and caregivers regarding their experiences and preferences related to the use of televideo technology for their neuro-oncology care.  The manuscript is well written with an appropriately detailed introduction and thoughtful discussion.  I think that the greatest strength of the paper is the presentation of the survey results, which gives a patient voice to share their values in telemedicine.  It is helpful to the reader to see that the patient experience largely matches the expert recommendations around the use of telehealth in neuro-oncology. 

My enthusiasm for the paper is diminished only because the use of video consultation is overall quite low in this cohort, and I do not come away with an understanding of why.  Of over 3700 consultations, only about 8% were conducted by televideo (313 visits).  114 patients chose to participate in one or more video consultation. Overall, despite a relatively positive experience with video consultation, the utilization was quite low.  It would be helpful to better understand this.  Did institutional policy influence the percent of visits offered by video vs in person?  Was patient choice/preference the driving factor?  Understanding these limitations would increase the generalizability of this data.

Author Response

We very much appreciate the opinion of the reviewer and their positive evaluation. We agree that one might suspect a higher number of participants in video consultations (VC).

Though the regulations on reimbursement by health insurance providers in Germany were unclear at the beginning of the pandemic, we aimed to offer the opportunity to participate in VC to all patients only by patients’ choice/preference and thereby without an institutional influence. That being said, due to the study’s retrospective nature, we can not definitively exclude such an influence.

We hypothesize that the low number of participants in VC might have been significantly influenced by the selected timeframe for our study and patients’ initial restraints/uncertainties.
We started VC at the beginning of the pandemic and consecutively included consultations beginning in April 2020 for this study. Up to October/November 2020, less than 5% of consultations were performed as VC each month. We speculate that this might be due to a general restraint of patients and their relatives towards the new concept and uncertainties regarding the technical configuration, as reported by multiple other investigators cited in the manuscript.
This theory is also supported by the increase in acceptance and participation over time, as seen in Figure 1a. In the spring of 2021, partially over 20% of consultations were held as VC. We assume that a closer investigation of VC in 2021/2022 would reveal significantly higher participation than observed in our investigative period.
Though we hope this hypothesis presents a satisfactory explanation for the reviewers’ questions, it is partially based on speculations, so we have not included it in the final manuscript. 

Reviewer 2 Report

Thank you for the opportunity to review this manuscript. Here, the authors analyzed over 3,700 neuro-oncological consultations, of which over 300 were conducted as video consultations (VC) by patients’ preference.

The study found that patients who participated in VC had worse clinical conditions, higher grades of malignancy, were more often diagnosed with glioblastoma, and had a longer travel distance.

VC was considered an adequate alternative to face-to-face consultations for almost all encounters patients chose to participate in (>70%), except initial consultations.

Most participants preferred to alternate between both modalities rather than participate in one alone but preferred VC over telephone consultation.

VC made patients feel safer, and participants expressed interest in implementing other telemedicine modalities (e.g., apps) into neuro-oncology.

Overall, the study provides valuable insights into the use of telemedicine in neuro-oncology and suggests that VC can be a promising addition to patient care.

Strengths are:

The study analyzed many neuro-oncological consultations (>3,700), increasing the statistical power and reliability of the findings.

The study included quantitative and qualitative data by analyzing consultation records and surveying patients about their experiences with video consultations (VC).

The study provides valuable insights into the use of VC in neuro-oncology by identifying suitable patient subgroups, occasions, and future directives for expanding and adapting telemedicine patient care.

The study’s findings may help to guide the selection of patients and encounter for VC in neuro-oncology in an individualized and evidence-based manner.

Limitations are: 

The study was conducted at a single institution, which may limit the generalizability of the findings to other settings.

The study relied on patients’ preference for participating in video consultations (VC), which may introduce selection bias.

The study did not compare the effectiveness of VC to face-to-face consultations or other telemedicine modalities.

The study did not assess the long-term outcomes or cost-effectiveness of using VC in neuro-oncology.

Compared to previous studies, the paper appears to provide new insights into using video consultations (VC) in neuro-oncology by analyzing many consultations and surveying patients about their experiences with VC. The study found that VC was considered a fully adequate alternative to face-to-face consultations for almost all encounters, except initial consultations, and that most participants preferred to alternate between both modalities. These findings may help to guide the selection of patients and encounters for VC in neuro-oncology.

Author Response

We highly appreciate the reviewer’s positive evaluation. As the reviewer mentions no points of concern or issues to improve, we have not made further changes to the manuscript.

Reviewer 3 Report

In this paper the authors retrospectively collect data on VC for neuro-oncological patients during CoViD pandemic (313 VC on 3751 total consultation), investigating patients and cargivers personal experience.

They conclude that VC for neuro-oncological patients seem to be a feasible option for clinical follow up session except for first evaluation. Patients reported a high degree of satisfaction on multiple domains.

The paper is well written and quite easy to understand. English is fine. Auto-citation rate is low (0/68). The topic fit the Special Issue.

Minor issue should be addressed:

-Results presentation should be more clear and understandable. Perhaps making more clear the subclassificiation (e.g. patient that fill the survey vs patients helped by the caregivers vs caregivers alone). Also one more scheme (see Figure 5) could be usefull.

-Carefull check for typo errors (see line 241, 316)

Author Response

"Minor issue should be addressed:

-Results presentation should be more clear and understandable. Perhaps making more clear the subclassificiation (e.g. patient that fill the survey vs patients helped by the caregivers vs caregivers alone). Also one more scheme (see Figure 5) could be usefull.”

We highly appreciate the reviewer’s overall evaluation of our manuscript and thank them for bringing these opportunities for improvement to our attention. We revised our materials & methods and results section, focusing on the subclassification mentioned above. We added an additional explanation of these subgroups in lines 132/133 combined with an abbreviation for each group that was added in the text whenever these groups were referenced in the manuscript. Furthermore, we revised Figure 3, adding a new Figure 3b that visually represents our significant findings regarding differences in the answers given in the survey depending on who completed it. Subsequently, we changed the numbering in the text.

Figure 3 revised
Figure 3. Reasons for and occasions of VC as stated by the participants in our survey. (a) Bar graph of participants agreeing/disagreeing with different reasons to choose VC over other modalities (multiple choice question). (b) Bar graph of participants agreeing/disagreeing with different reasons to choose VC over other modalities (multiple choice question) divided by the persons completing the survey: Patients themselves (P, blue), patients together with their relatives (PR, orange) or relatives alone (e.g. if the patients were unable or deceased, R, black). (c) Bar graph of occasions that participants stated to have experienced a VC in reality (blue) and considered a fully adequate alternative to face-to-face consultations irrespective of whether they experienced a VC for this encounter in reality (green, multiple choice questions). (d) Participants’ statement of for which occasions where they considered VC to be a fully adequate alternative only including patients/relatives that did in reality experience a VC for the respective occasion (e.g. percentage of patients that considered control visits after the MRI control to be a fully adequate alternative via VC of the collective of patients that experienced at least one control visit after the MRI control as VC).

“-Carefull check for typo errors (see line 241, 316)”

We apologize for our oversights and have performed additional manual and software-based language checks for typos and other errors.